

# Reproductive management: conditioning, spawning and development of Peruvian grunt *Anisotremus scapularis* in southern Peru

Renzo Pepe-Victoriano[1,2], Jordan I. Huanacuni[3,4], Pablo Presa[5] and Luis Antonio Espinoza-Ramos[6]

[1] Facultad de Recursos Naturales Renovables, Área de Biología Marina y Acuicultura, Universidad Aturo Prat, Iquique, Iquique, Chile
[2] Núcleo de Investigación Aplicada e Innovación en Ciencias Biológicas, Facultad de Recursos Naturales Renovables, Universidad Arturo Prat, Arica, Arica y Parinacota, Chile
[3] Facultad de Recursos Naturales Renovables, Universidad Arturo Prat, Arica, Arica y Parinacota, Chile
[4] Finfish Aquaculture Sociedad Anónima Cerrada, Tacna, Tacna, Peru
[5] Laboratory of Marine Genetic Resources (ReXenMar), CIM, Universidad de Vigo, Vigo, Spain
[6] Facultad de Ciencias Agropecuarias, Escuela Profesional de Ingeniería Pesquera, Universidad Nacional Jorge Basadre Grohmann, Tacna, Tacna, Peru

Corresponding author
Luis Antonio Espinoza-Ramos, lespinozar@unjbg.edu.pe

## ABSTRACT

The Peruvian grunt, *Anisotremus scapularis*, is beginning its domestication as a candidate species for marine aquaculture. The optimal management of fingerling production requires precise knowledge on early development. Herein, we report the methodology for capturing and conditioning wild specimens to find a viable broodstock. The speed of capture and transportation (about 30 min), the post capture preventive treatment (60 min with tetracycline), and the 6-days preventive antiparasitic treatment (29 ppm formalin) maximized survival and a rapid feeding adaptation. Progressive diets based on the copepod *Emerita analoga*, fish meal, pellets and processed feedstuff prompted the spontaneous broodstock spawning 7 months post-capture. The interannual spawning of this broodstock since September 2016 indicated the optimal control of its reproduction in captivity. The morphogenetic process of the embryo lasted 42 h at 18 °C compared to (31–41) h at 19 °C in northern Peruvian latitudes. The knowledge generated allowed us to work out broodstock and egg management protocols in southern Pacific latitudes (southern Peru and northern Chile). Such protocols would help to escalate larval and juvenile production and to alleviate fishing pressure on the overexploited Peruvian grunt population.

# INTRODUCTION

The Peruvian grunt, *Anisotremus scapularis*, is a carnivorous marine fish with benthophagous habits that forms large fish schools in the eastern South Pacific, *i.e.*,

Ecuador, Peru and Chile (*Froese & Pauly, 2019*). The species shows a coastal distribution in rocky intertidal and subtidal zones (*Berrios & Vargas, 2004*) at moderate depths (*Carrillo Estévez, 2009*; *Méndez-Abarca & Pepe-Victoriano, 2020*). The high commercial value of this grunt in the artisanal fisheries of Peru (*Medicina, 2014*) and Chile (*The Nature Conservancy, 2020*) makes it a candidate for marine aquaculture. A crucial technological advancement has been carried out in the last decade to gauge the critical reproductive process and assess the feasibility of its cultivation. In the first experience carried out in the Morro Sama cultivation center (FONDEPÉS, IMARPE, Tacna, Peru), a system for capturing and conditioning wild specimens was designed (*Espinoza-Ramos et al., 2019*; *Huanacuni & Espinoza-Ramos, 2019*). Subsequently, significant advances were made *in vitro* in the biology of embryonic and larval development of the species (*Montes et al., 2019*). Also, the effect of fry culture density on growth, feeding efficiency and survival was evaluated (*Espinoza-Ramos et al., 2022a*). Recently, *Espinoza-Ramos et al. (2022b)* identified the effect of population density on growth, feed efficiency, and survival during the transition from Peruvian grunt fry to juveniles. Also in the last year, the study of the reproductive cycle has been completed as well as the spawning dynamics in the central coast of Peru, which provide important advances to understanding the reproductive cycle of females in this species (*Carrera Santos et al., 2022*). At a population scale, the mitogenome of this seabream has been developed (*Gomes-dos-Santos et al., 2020*), which constitutes a valuable tool to carry on studies on ecology, evolution, and conservation of this appreciated marine resource.

The development of studies aiming to determine the cultivation feasibility of this candidate species require more stress on both, its reproductive strategy in captivity (*Aydin, Polat & Sahin, 2020*; *Gomathi et al., 2020*) and its embryonic and larval development (*Pepe-Victoriano et al., 2021a*). For instance, many attempts on commercial escalation of aquaculture candidate species have consistently failed in the last decades because of viability problems in early stages of culture (*Boyd et al., 2020*). Therefore, precise knowledge on embryonic development under controlled conditions constitutes the biological basis for developing captive breeding protocols for candidate species (*Pepe-Victoriano et al., 2022*). Furthermore, success in aquaculture production of any marine fish has two additional requirements. The first is the establishment of a broodstock from wild adults, which can be conditioned to spawn in captivity (*Flores & Rendíc, 2011*; *Stieglitz et al., 2017*). Also, it is known that the organogenesis of pelagic eggs from marine teleosts depends on local oceanographic conditions (*Botta et al., 2010*; *Pepe-Victoriano, Araya & Faúndez, 2012*; *Marancik, Richardson & Konieczna, 2020*; *Ahmad Syafiq, Mohamad & Mohidin, 2020*; *Aoki et al., 2020*; *Oka et al., 2020*). In this regard, in addition to the advances achieved in the embryology of this species in central Peru (*Espinoza-Ramos et al., 2022a*), it is important to generate information on its embryonic development in other latitudes of the South Pacific. Therefore, the second goal is to gauge the geographic variation of embryonic development, which is decisive for designing local aquaculture management protocols. That knowledge would also be useful in recognizing embryos and

larvae in natural environments for assessment purposes (*Gisbert, Piedrahita & Conklin, 2004*; *Botta et al., 2010*; *Betti, 2011*; *Pepe-Victoriano, Araya & Faúndez, 2012*; *Marancik, Richardson & Konieczna, 2020*; *Oka et al., 2020*).

This work seeks progress on the management of the Peruvian grunt in southern Peru, where its cultivation can alleviate the unsustainable overfishing of this species (*Allen, 2001*). In addition, the development of its aquaculture would represent an important input for the depressed economy of the semi-desert region of southern Peru and northern Chile. The objective of this study is to establish the protocol for capture, transport and maintenance of wild adults of *A. scapularis* in an open culture system, which allows for reproductive maturation, spawning and embryonic development in southern Peru.

## MATERIALS AND METHODS

The research was carried out from March 2016 to May 2017 at the Morro Sama Aquaculture Center (CAMOSA) of the National Fisheries Development Fund (FONDEPES), (17°59′39.7″S, 70°52′59.1″W), in the Peruvian region of Tacna. All procedures and animal handling described in this study were carried out according to the Peruvian law #30407, "Law on Animal Protection and Welfare" governed by The Ministry of Agriculture and Irrigation (MINAGRI) (*El Peruano, 2016*) and the ethical review of research protocols and their supporting documents was carried out by Scientific Ethics Committee—Aquainnova Ltda (Res. 009-2024).

### Capture and transport of *A. scapularis*

The fishing area included the rocky beaches of Llostay in Tacna, Peru (18°10′26″S, 70°38′37″W) (Fig. 1). Three fishing campaigns were carried out during the morning and afternoon on March 11 and 17 and on April 6 of 2016, in which 15 *A. scapularis* adults were captured (Fig. 2). The campaigns were carried out with fishing rods with BaitHolder hooks at 50 m offshore from the low tide line, because this capture method minimizes stress through careful handling and rapid transport, using *Emerita analoga* (Pacific sand crab) as fishing bait. The fishes were moved to the beach shore, the hooks were removed by expert fishermen, and the specimens were stabilized in a 50 L oval tub using jets of whipped seawater using a 20 L bucket and 2 L jugs. The weight of the specimens was recorded *in situ* with a digital scale (0.1 precision, Makita, La Mirada, CA, USA) and the size was measured with an ichthyometer (1 mm precision). The transference of specimens to the aquaculture centre comprised less than seven individuals per tank or 3 kg of live weight. The transports were driven from Llostay Beach to the Morro Sama aquaculture center (17°59′39.7″S, 70°52′59.1″) in a 1 m³ fiberglass tank filled to 500 L with raw seawater installed in a 4 × 4 truck. The oxygen supplied through a silicone hose ending in an air stone maintained the $O_2$ concentration saturated at (90–98)% in the transport tank. The temperature during the transport was maintained between 19 °C and 20 °C by means of 1 kg ice bags immersed in the tank.

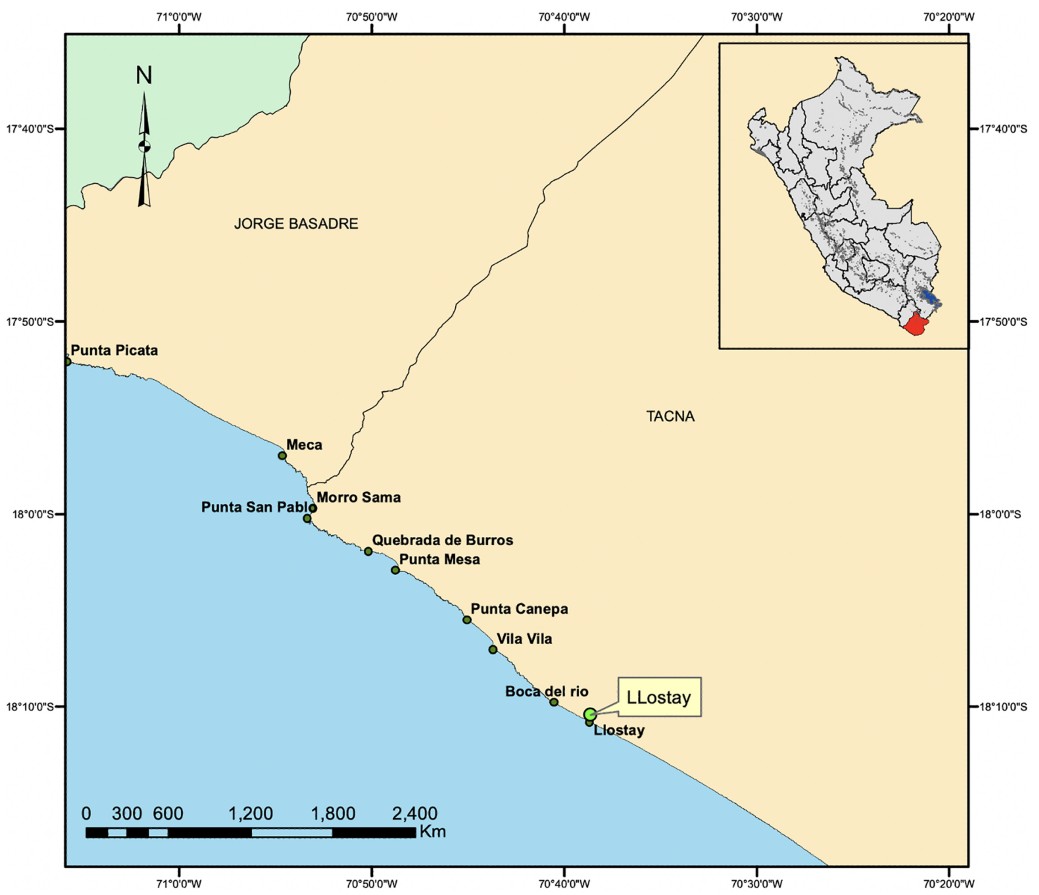

**Figure 1 Capture area of *A. scapularis* in Llostay Beach, southern Peru.**

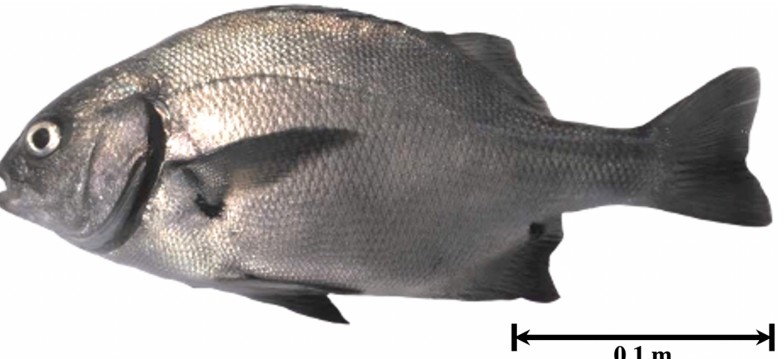

**Figure 2 Specimen of *Anisotremus scapularis* from southern Peru (Peruvian grunt, seabream).**

## Conditioning and feeding

At the Morro Sama aquaculture center, a preventive treatment with 99% oxytetracycline was applied at 50 ppm for 60 min in the transport tank to prevent infections on putative injuries caused by fishing/handling. The conditioning stage lasted seven days during which

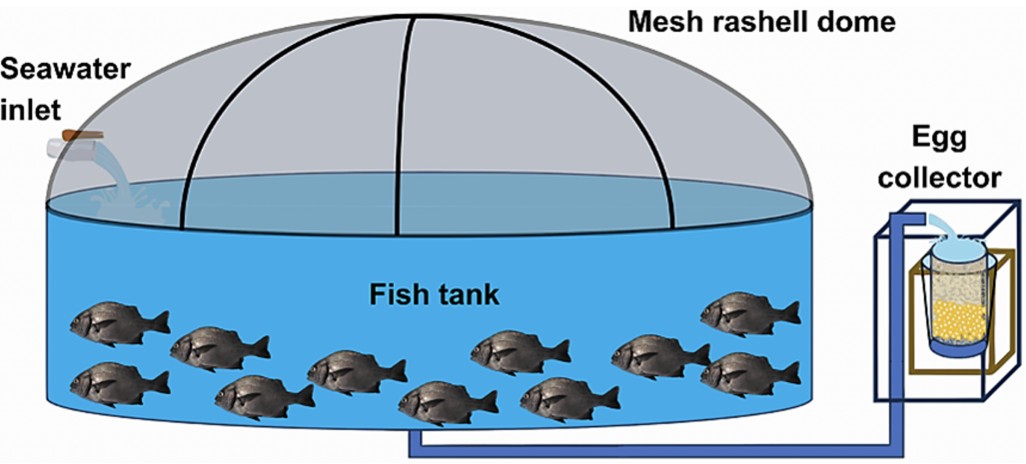

**Figure 3 Indoor pilot culture pond designed for *A. scapularis*.**

preventive antiparasite treatments were carried out with 37% formalin, *i.e.*, 1 ml of formalin per 6 L of water (23.6 ppm). The open culture system consisted of an Australian-type circular conditioning tank (Fig. 3) made of corrugated steel plates covered with a black PVC geomembrane. Its volumetric capacity was 9 m$^3$ (4 m in diameter and 1 m in height) without a filter system, water inlet, and outlet by constant pumping at 0.61 L/seg. Tank cleaning was performed by emptying and sweeping the tank twice a week (Tuesday and Friday) in the morning (8 am). Physicochemical parameters were recorded daily at 2:00 h, 8:00 h, 14:00 h and 20:00 h. Water parameters such as temperature, dissolved oxygen concentration and pH were measured with a thermometer, an oximeter (YSI model 550) and a pH-meter (ECOSENSE-pH100A pH-meter; YSI, Yellow Springs, OH, USA), respectively. Air inflow to the fish tank was regulated to maintain the dissolved oxygen concentration at 5.0–7.0 mg/L. The concentration of ammoniacal nitrogen (N-NH$_3$), ammonia (NH$_3$) and ammonium (NH$_4$) were determined monthly with a DR-400 HACH spectrophotometric kit (range 0.000–2.500 mg/L). Salinity was measured once a week with an RHS-28ATC refractometer.

Feeding the broodstock consisted of progressive diets based on the copepod *E. analoga*, fish meal, pellets and processed feedstuff. During the first post-treatment stage (38 days from March 18 to April 25, 2016), the frozen Pacific sand crab *E. analoga* was supplied daily *ad libitum* at a rate of 5% of total biomass. The subsequent diet was applied for 34 days (April 26 to May 24, 2016) and consisted of coating the *E. analoga* in fish meal, at a rate of 4.06% of total biomass. Starting in June 2016, the definitive diet consisted on an artisanal-made semi-moist feed (*i.e.*, its proximal analysis gave 37.13% protein, 6.57% fat, 9.45% ashes, 0.73% fiber, and 44.32 humidity, (*Montes et al., 2019*)) delivered twice a day at a rate of 2.24% of live biomass and combined with whole frozen *E. analoga* two times per week, to make it attractive and palatable to the Peruvian grunt specimens. On October 4, 2016, an intramuscular individual identification chip (FELIXCAN-FDX-B) was implanted in the dorsal area of the specimens to facilitate individual monitoring of growth and

**Table 1 Production and growth values of *A. scapularis* broodstock in period October 2016–May 2017.**

| Variable | Broodstock tank |
| --- | --- |
| Food provided (kg) | 21.98 |
| Initial biomass (kg) | 10.85 |
| Final biomass (kg) | 13.79 |
| Increase in weight (Kg) | 2.94 |
| Initial density (Kg/m$^3$) | 0.69 |
| Final density (Kg/m$^3$) | 0.88 |
| Initial No. of fish initial | 11 |
| Final No. of fish | 11 |
| FCR | 7.46 |
| SGR | 0.13 |
| Weight gain (%) | 27.16 |
| Survival rate (%) | 100 |

**Note:**
SGR, specific growth rate; FCR, food conversion rate.

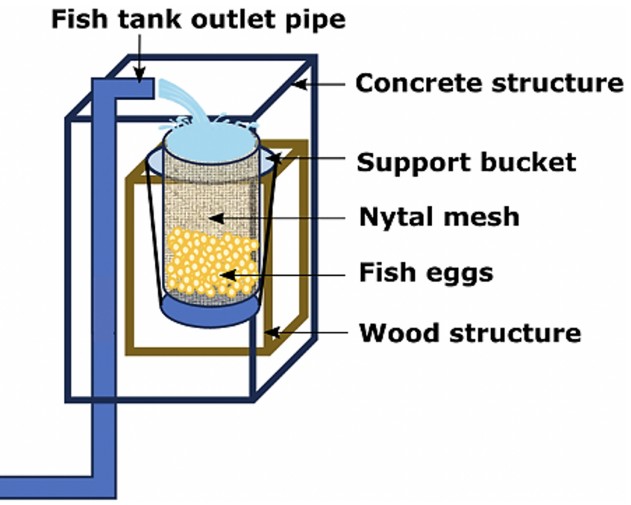

**Figure 4 Egg collector designed for *A. scapularis* and placed at the tank edge.**

spawning. The values of FCA, SGR and the percentages of growth and survival are presented, together with other data obtained in the different samples (Table 1).

## First spawning and incubation of eggs

Only adult fishes above 500 g in weight were captured to fund an aquaculture broodstock. The authors used the biometric manipulations to gauge their sex and maturity status throughout abdominal massage. The eggs were systematically collected upon laying in a cylindrical collector of 300 μm planktonic mesh, with a circular ring at the top that fit into a rigid vertical frame of 0.3 × 0.3 × 0.6 m, placed at the outlet of the tank overflow (Fig. 4). The eggs were transferred to a 20 L bucket and transported to the incubation room where

they were washed on a 300 μm sieve with sterilized seawater filtered at 1 μm in a cartridge filter and treated with UV (AL-PVC -160W). The eggs were then transferred to a 1 L test tube in sterile water for 10–15 min to separate the dead eggs precipitated at the bottom. Finally, the volumetric count of the floating eggs (fertilized and viable) and the precipitates (non-viable) was carried out. The viable laying fraction was immersed for 5 min in 4 L of a 1.5% Aqua-Yodo dilution at a rate of 2 mL/L of seawater. Subsequently, they were kept for 4 days in a black fiberglass tank of 500 L capacity with 400 L seawater (salinity range 34.7–35.6 psu and temperature 17.36 ± 1.54 °C), with a light supply of air through stone diffusers (air stone model ASI-3) at a rate of 0.1 L/min. This system allowed the uniform distribution of the eggs in the water column at an incubation density of 50 egg/L. The partial replacement of 30% seawater was carried out daily (9:00 a.m.) in the incubation tank using a 300 μm sieve. This protocol for egg management was carried out after each spontaneous spawning.

## Egg sampling and morphological characterization of the embryo

In each egg sampling, the temperature and hydrogen potential were recorded. From the first day of incubation until hatching, a per-hour subsampling of 130 mL water was randomly taken to characterize the embryonic development according to *Montes et al. (2019)*. Eggs were observed and photographed under a stereoscopic microscope (Japan Optical Co. Ltd. model XTL-2310, Shiga, Japan) and their characteristics were documented.

## Statistical analysis

Handling of specimens required the use of eugenol (106 mg/L) as anesthesia. From April 2016 to October 2016, monthly biometrics were carried out with a 1 mm precision ichthyometer to gauge size growth of the broodstock. The weight was recorded on a 5 kg scale (Coretto EC-30). The distribution of these parameters was normal, and correlations (Pearson) were carried out with the statistical package R-studio version 1.4.1106 (R-studio, Inc., Boston, MA, USA), the ggplot2 package and the Excel package. The specific growth rate SGR was calculated according to *Cook et al. (2000)* using the following equation, either for the weight or the body length in terms of gain percentage per day.

Specific growth rate (SGR),

$$SGR = (Lf\ Wf - Li\ Wi/t) \times 100 \tag{1}$$

where SGR is the specific growth rate being Wf and Wi, the final and the initial fresh body weight (in grams), respectively, Lf, Li, the final and the initial body length in centimeters, respectively, and t the period between the initial and the final feeding period measured in days.

The viability rate was calculated for each spawning as,

$$\text{Viability rate (\%)} = (\text{Floating eggs/Total No. eggs}) \times 100. \tag{2}$$

The hatching rate was determined after 48 h of incubation as,

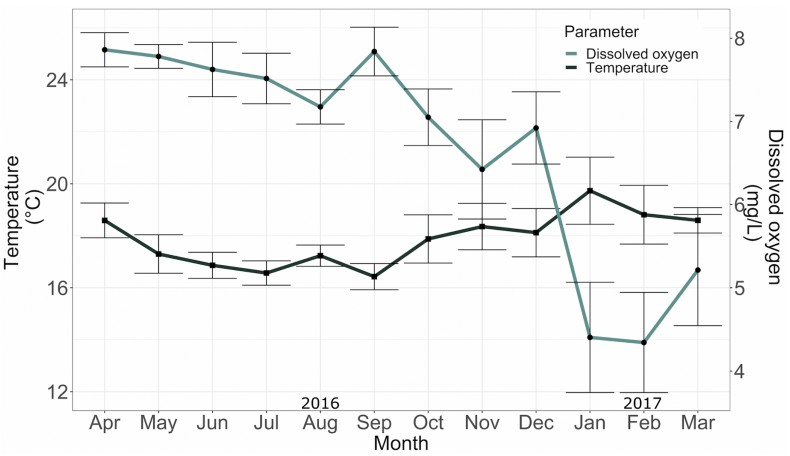

**Figure 5 Temperature (°C) and dissolved oxygen (mg*L⁻¹) in the conditioning tank for adult specimens in 2016.**

$$\text{Hatching rate (\%)} = (\text{Hatched Larvae/Floating eggs}) \times 100. \tag{3}$$

Ninety-six hours after the start of egg incubation, the number of larvae with an absorbed yolk sac was counted to determine larval survival as,

$$\text{Survival rate (\%)} = (\text{Larvae with yolk sac absorbed/Hatched larvae}) \times 100. \tag{4}$$

# RESULTS

## Capture and transport of *A. scapularis*

The seawater temperature ranged from 18 °C to 18.5 °C during the Peruvian grunt fishing campaigns in March and April 2016. The global survival rate of the captured fish in the three campaigns ranged from 0% to 46.7%. Total survival amounted to 73.33% (11 specimens) and mortality reached 26.67% (four specimens). The latter specimens did not survive due to physical damage caused by the fishing gear, which consisted of fatal injuries in mouth and gills. Eight males and three females were identified with lengths ranging from 30.0 to 44.7, with weights ranging from 0.598 to 2.293 kg. The density of fish transported from Llostay Beach to the Morro Sama aquaculture center ranged 4–7 fish/m³ and a maximum of 3 kg of live weight per tank. No correlation was observed between the number of fish transported per tank and survival, *i.e.*, transport densities of 4 fish/m³ resulted in a survival rate of 26.7% but 7 fish/m³ resulted in a 46.67% survival rate.

## Conditioning and feeding

The mean temperature and mean dissolved oxygen in the conditioning tank between April 2016 and May 2017 were 17.36 ± 1.54 °C and 5.96 ± 1.03 mg/L, respectively (Fig. 5).

The chemical parameters showed concentrations of 17 ± 0.38 µg/L ammoniacal nitrogen (N-NH₃), 1.21 ± 0.02 µg/L ammonia (NH₃) and 1.29 ± 0.03 µg/L ammonium (NH₄). The salinity ranged 34.7–35.6 psu (Fig. 6) and the pH (± SD) was 7.29 ± 0.63.

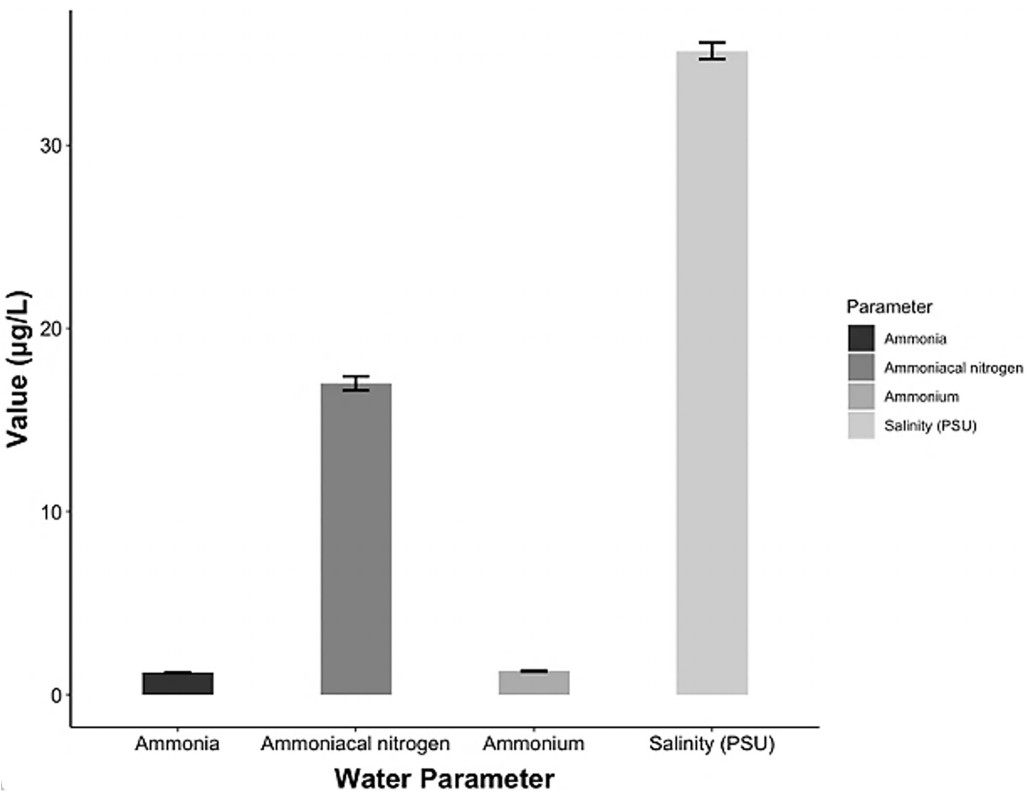

Figure 6 Chemical parameters in the conditioning pond for the *A. scapularis* broodstock.

Table 2 Per month weight, length and total biomass of the Peruvian grunt broodstock in 2016.

| | Average weight (kg/specimen ± SD) | Average size (cm/specimen ± SD) | Total biomass (kg) |
|---|---|---|---|
| April | 0.986 ± 0.470 | 34.91 ± 5.367 | 10.845 |
| May | 1.057 ± 0.506 | 35.24 ± 5.331 | 11.632 |
| June | 1.139 ± 0.529 | 35.64 ± 5.158 | 12.524 |
| July | 1.199 ± 0.544 | 35.96 ± 5.107 | 13.186 |
| August | 1.249 ± 0.567 | 36.79 ± 5.167 | 13.690 |
| September | 1.231 ± 0.556 | 37.13 ± 5.104 | 13.540 |
| Inc./month | 0.049 + 0.037 | 0.444 ± 0.202 | 0.539 ± 0.411 |

The 11 live specimens of Peruvian grunt conditioned as broodstock accepted the preventive treatment as well as the nutrition program ending by semi-moist balanced food with 37% protein at a feeding rate of 2.24% combined with the copepod *E. analoga*. The increase in average weight of the spawners biomass in the period between April and September 2016 was 539 ± 411 g/month, their average individual weight gain was 0.049 ± 0.037 g/month, and the average individual length increase was 0.444 ± 0.202 cm/month (Table 2). The correlation of weight-length increments per month was significant and positive (Pearson, $r = 0.968$, two-sided $p = 0.001$) (Fig. 7).

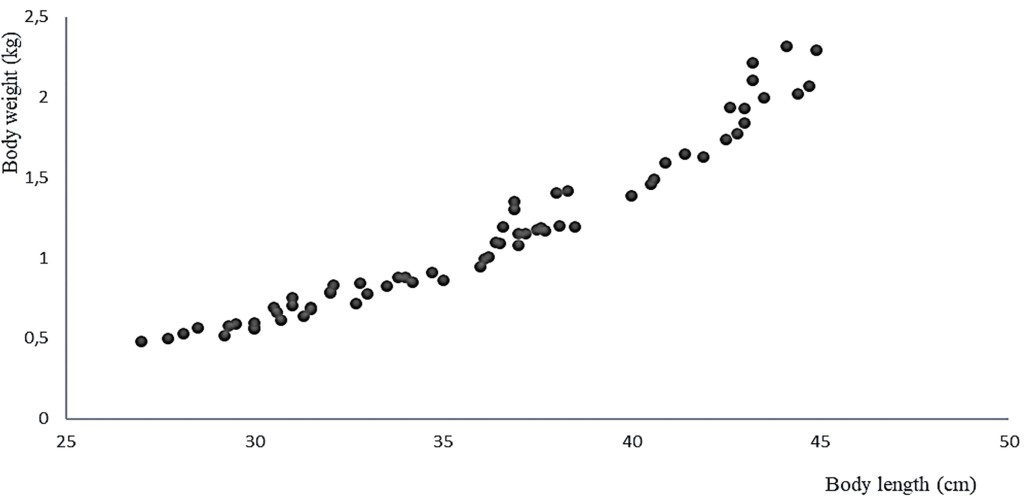

**Figure 7** Relationship between body weight and body size of the *A. scapularis* broodstock from April to September 2016 (Pearson *r* = 0.968, *n* = 66, *p* = 0.001).

## First spawning of *A. scapularis* broodstock

After 7 months under indoor conditioning (from October 2016 to May 2017), the broodstock spawned daily between 5:00 p.m. and 7:00 p.m. in a temperature range of 18–20 °C, natural light intensity of 335.85 lux and water flow rate of 0.61 L/s. Sex determination was affordable during biometric controls when the abdominal massage allowed to verify the sex of the expelled gametes. Eight and three out of 11 grunt specimens were males and females, respectively, and the sex ratio in the reproduction tank was ~3:1 male:female.

The dynamics of spontaneous spawning of *A. scapularis* in captivity from October 2016 to May 2017 showed two main peaks, in December–January and in March (Fig. 8).

Metrics for the spawning activity are given in Table 3. The survival rate indicates the percentage of viable eggs collected within 24 h post spawning (hps). Egg hatching (larvae with yolk sac) begun at 48 hps, the hatching rate was confirmed at 72 hps and larvae survival (larvae with or without the yolk sac consumed) was scored at 96 hps. All hatching parameters were significantly lower in October–November 2016 (beginning of spawning) and in February (middle spawning season). Hatching rate and larvae survival rate did not differ significantly throughout the spawning season (Table 3).

## Embryological development of *A. scapularis*

The average diameter of Peruvian grunt oocytes was 735 ± 0.113 μm, that of fertile eggs was 800 ± 0.207 μm and the hatched larvae had 1.87 ± 0.05 mm in length. The main phases of the embryo development were completed at 42 hpf (hours post-fertilization) (Table 4).

The viable eggs were spherical, non-adherent and telolecithal type, *i.e.*, they exhibited a large amount of translucent yolk at the vegetal pole and a discoidal meroblastic division during the first cleavages (Fig. 9). At 0.35–0.45 hpf the zygote was translucent, contained oily droplets in its center and presented a conspicuous accumulation of cytoplasm towards the animal pole to form the blastodisc (Fig. 9A). Between 1.00–2.32 hpf, the blastodisc

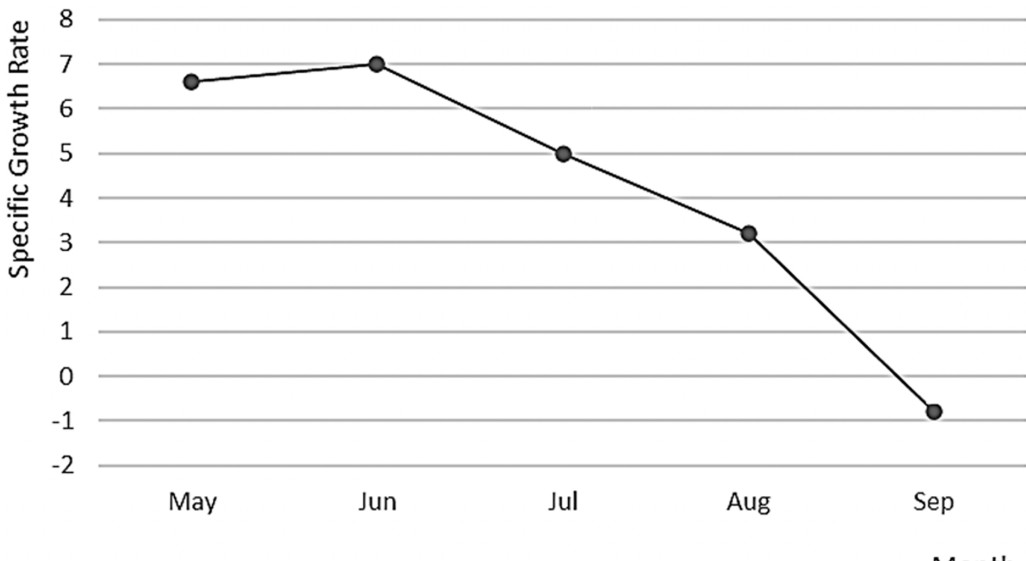

**Figure 8 Specific growth rate of *A. scapularis* broodstock from April to September 2016.**

**Table 3 Dynamic properties of spontaneous spawning of the first *A. scapularis* broodstock in captivity in period 2016–2017.**

| | | No. batches | No. egg (× 10⁶) | Survival rate (%) | No. fertilized eggs (× 10⁶) | Egg hatching (× 10⁶) | Hatching rate (%) | Larvae survival (%) |
|---|---|---|---|---|---|---|---|---|
| 2016 | October | 2 | 0.132 | 75.11 | 0.099 | 0.06 | 64.94 | 73.62 |
| | November | NS | – | – | – | – | – | – |
| | December | 15 | 6.776 | 97.39 | 6.415 | 4.70 | 71.25 | 55.60 |
| 2017 | January | 25 | 6.903 | 92.68 | 6.308 | 4.67 | 73.15 | 56.34 |
| | February | 4 | 0.368 | 50.86 | 0.187 | 0.15 | 82.97 | 54.06 |
| | March | 20 | 5.208 | 87.35 | 4.550 | 3.52 | 77.38 | 72.85 |
| | April | 16 | 2.114 | 83.80 | 1.772 | 1.43 | 80.86 | 71.13 |
| | May | 12 | 1.777 | 86.61 | 1.508 | 1.35 | 87.93 | 67.93 |
| | Mean | 13.43 | 3.330 | 81.97 | 2.936 | 2.27 | 76.92 | 64.50 |
| | ±SD | 8.24 | 2.920 | 15.39 | 2.970 | 2.01 | 7.78 | 8.79 |

**Note:**
NS, No spawning activity was observed.

giving rise to the first blastomere had already formed. The division of the blastomeres was symmetrical, the daughter cells were like each other in size (second cleavage). The development continued with the loss of symmetry in the blastomeres and their progressive decrease in size with further divisions (third cleavage). As in the previous step, there was no difference in cell shape and size at the 16 blastomeres stage (fourth and fifth cleavages; Fig. 9B). Between 4.55–5.60 hpf distal cells were smaller than their adjacent ones. A morula was formed with two cell layers. There was a large association of small cells from irregular to circular shape and flattened dorso-ventrally, concentrated in a single cell pole (Fig. 9C).

**Table 4 Morphological characterization of development stages in *A. scapularis*.**

| Stage | General characteristics | (h:min)pf | Figure |
|---|---|---|---|
| Zygote | Fertilized egg. | 0.35–0.45 | 1A |
| First cleavages | 32 Blastomeres. | 1.00–2.32 | 1B |
| Morula | 512 cell stage. | 4.55–5.60 | 1C |
| Blastula | Blastopore, blastocele and onset of epiboly. | 13.0–14.0 | 1D |
| Gastrula | Neural groove and germ ring. | 15.55–18.50 | 1E |
| Organogenesis | Head region, optic vesicles, beating heart, somites and caudal pedunculus. | 20.26–40.00 | 1F |
| Full embryo | Complete organogenesis and initial chorion breakdown. | 41.50–42.20 | 1G |
| Hatching embryo | Embryo movements and dissolution of the chorionic membrane. | 42.15–42.20 | 1H |
| Hatched larva | External yolk sac, oil droplet and non-functional mouth. | 42.25 | 1I |

**Note:**
(h:min)pf: hours and minutes post fertilization.

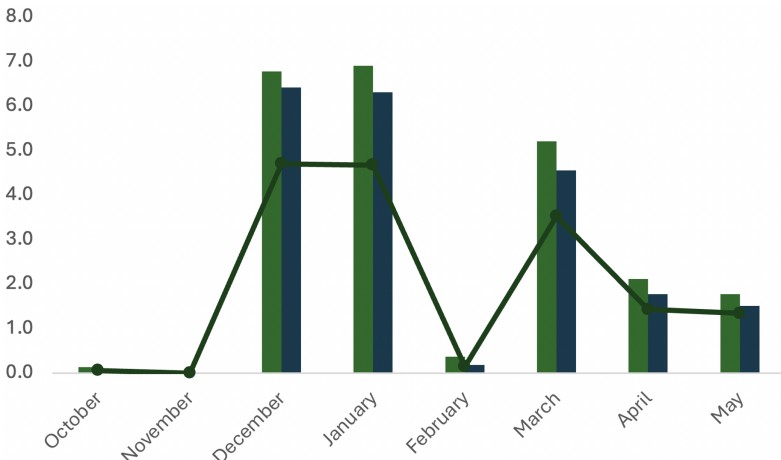

**Figure 9 Dynamic properties of spawning of the first *A. scapularis* broodstock in captivity from southern Peru, from October 2016 to May 2017.** No. eggs hatched ($\times 10^6$, line); No. fertilized eggs ($\times 10^6$, dark blue), No. eggs spawned ($\times 10^6$, grey bar).

Between 13.0–14.0 hpf, the blastopore as origin of the blastocele was observed. The cell movement (epiboly) of blastomeres and the extension of the cell mass through the egg in the direction of the equatorial pole begun. The peripheral dendriform cells adopted the shape of a translucent rim (Fig. 9D). Between 15.55–18.50 hpf the blastoderm had covered approximately 1/4 to 1/3 of the yolk sac, the area where the neural groove appears was already identifiable and the germ ring begun to form (Fig. 9E). Between 20.26–40.0 hpf, the cephalic region was differentiated from the caudal region, and was still flattened dorso-ventrally due to the presence of the ocular buds. The cephalic region expanded laterally, and the optic lenses and optic vesicles were distinguished. A two-chambered beating heart was observed, and the closure of the blastopore gave rise to the Kuffper vesicle. The growth of the tail was observed in the perivitelline space, and the size of the embryo exceeded the amplitude of the yolk sac. The tip of the tail was levelled with the eyes and the thickening of the body was observed in a dorso-ventral direction. The somites were

well defined, and the embryo begun moving and a duct was observed between the notochord and the heart, which corresponded to the digestive tract. In the caudal terminal region, a narrowness corresponding to the caudal pedunculus was observed and there was a marked development of the face (Fig. 9F). Organogenesis was completed between 41.50–42.15 hpf, the chorion began to tear, and an incipient ocular pigmentation was observed (Fig. 9G). Between 42.15–42.20 hpf, the embryo performed a series of spiral contortions and the chorionic membrane dissolved in less than 5 min (Fig. 9H). About 42.25 hpf, a newly hatched larvae was observed with a single embryonic fin which extended from the cephalic region to the rostral region. The yolk sac covered half the total length of the larva, and a closed, non-functional mouth was observed (Fig. 9I).

## DISCUSSION

Achieving the capture, acclimatization, feeding, spontaneous spawning of a broodstock and characterizing the embryological development are essentials for the domestication of new species (*Pepe-Victoriano et al., 2022*). The control of reproduction has allowed the production of several species that were initially far from domestication (*Wexler et al., 2003*; *Flores & Rendíc, 2011*; *Bar et al., 2015*). Building a broodstock through wild capture requires handling minimization to prevent harmful stress manifested by physiological alterations (*Carrera Santos et al., 2018*; *Espinoza-Ramos et al., 2019*). The capture of *A. scapularis* carried out between March and April 2016 showed that the protocol applied in surf zones resulted in low handling stress, which could have positively affected the subsequent good acclimatization in captivity. A low stress in this species has also been observed in other capture and transport systems (*Vollmann-Schipper, 1978*; *Espinoza-Ramos et al., 2019*) and is believed to be due in part to its lower requirements with respect to pelagic species, *e.g.*, Scombrids, which need to swim constantly due to negative buoyancy and ram-type ventilation (*Haux, Sjöbeck & Larsson, 1985*).

The low temperature maintained during transport seems to reduce the metabolism and stress of the fishand to prevent mortality due to oxygen deficiency (*Vollmann-Schipper, 1978*; *Pepe-Victoriano et al., 2022*). The weight and putative age of the captured fish might have also contributed to reduce stress, *i.e.*, less than 1 kg/fish, as recommended for other species (*Wexler et al., 2003*). The low transport density (maximum of 4–7 fish and 3 kg fish per tank) also guaranteed a high survival and optimal adaptation to culture conditions, as shown in *Thunnus albacares* (*Wexler et al., 2003*), *Euthynnus affinis* and *Cybiosarda elegans* (*Bar et al., 2015*) or *Sarda chiliensis chiliensis* (*Pepe-Victoriano et al., 2021b*, *2021a*, *2022*). In some fishes, the physiological parameters affected by the stress of handling, recover and stabilize after 2–4 days post-capture, *e.g.*, *Perca fluviatilis* (*Haux, Sjöbeck & Larsson, 1985*; *Huntingford et al., 2006*; *Davis, 2010*). However, post-traumatic stress can lead to lack of appetite and high early mortality (*Bar et al., 2015*). The punctual mortality observed in *A. scapularis* four days after starting his conditioning period was due to internal injuries to the mouth and gills caused by the capture. However, no subsequent mortalities were observed during conditioning, which is explained by low handling stress as well as adequate preventive treatments and nutrition.

Long-term chronic stress reflected by physiological changes can be caused by confinement, starvation, lighting, thermal stress and tank size as reported in *Clarias gariepinus* (*Aiyelari, Adebayo & Osiyemi, 2007*). Immediate transport after capture and a rapid conditioning minimized stress and allowed a rapid adaptation to feeding in captivity (*Ortega & De la Gándara, 2007*). For example, the provision of fresh or frozen diets has previously been carried out successfully for conditioning a wild broodstock of *Paralichthys adspersus* (*Muñoz, Segovia & Flores, 2012*) and *Graus nigra* (*Silva & Oliva, 2010*). In the Peruvian grunt, we have tested a successful progressive diet consisting on fresh or frozen *E. analoga*, followed by a mixture of this copepod and dry pellet, and ending up with semi-moist pellet composed by fish meal and *E. analoga* (see also alternative diets (*Carrera Santos et al., 2018*)). This protocol was different from those used in marine aquaculture where feeding is normally based on live, fresh and/or fresh frozen fishes (*Yazawa et al., 2017*; *Pepe-Victoriano et al., 2022*), and led to achieve maturation and a good spontaneous spawning production.

The Peruvian grunt has been described as a moderate to slow-growing species capable of reaching 15.46 ± 1.01 cm in length (*Instituto del mar del Perú: IMARPE, 2015*) and approximately 0.2 kg in weight in the first year (*Dionicio-Acedo et al., 2018*). This study is the first carried out on reproductive adults of this species and confirms that the highest increase in reproductive biomass occurs between April and August (maximum SGR in May 2017). On the contrary, the lowest growth rate was observed during the spawning season, *i.e.*, between October 2016 and March 2017 (*Barrett, 1971*; *Goldberg & Mussiett, 1984*). There were two main picks of higher spawning activity and larvae survival, in December–January and March–April, with February as a recovery period between picks (see Fig. 10). This spawning regime allows us to suggest that the Peruvian grunt is a synchronous batch spawner.

The eggs analyzed from different batches were telolecithal type with a discoidal meroblastic cleavage pattern and the animal pole was restricted to a small area (*Hall, Smith & Johnston, 2004*; *Falk-Petersen, 2005*; *Montes et al., 2019*). The embryonic development of this species was similar to previous reports on other marine teleosts (*Nande et al., 2017*), *e.g.*, the average diameter of Peruvian grunt eggs was 735 ± 0.113 μm, which coincides with previous estimates under laboratory conditions (0.752 ± 0.025 mm (*Montes et al., 2019*)) and also with the egg diameter in the black grunt *Haemulon bonariense* (0.80 ± 0.05 mm, (*Cuartas et al., 2003*)), the red snapper *Lutjanus campechanus* (0.82 mm; (*Papanikos et al., 2003*)), the yellow snapper *Lutjanus argentiventris* (0.75 mm; (*Muhlia-Melo et al., 2003*)) and the oxeye snapper *Lutjanus russellii* (0.71–0.84 mm (*Leu & Liou, 2013*)). Recently *Cota Mamani et al. (2023)* reported that the average egg diameter for *A. scapularis* is slightly larger (0.81 ± 0.07 mm). It has been shown that the inter-stage variability in the development time depends on culture temperature, with the initial and final phases of this process being faster than the intermediate ones. Such delay in the intermediate stages seems to be due to the process of blastopore closure, which leads to an increase in the total incubation time (*Le Clus & Malan, 1995*). In this regard, two main differences between previous reports on the embryonic development of this species are remarkable. First, hatching occurred at 42 hpf at 18 °C but between 31–41 hpf at 19 °C in laboratory

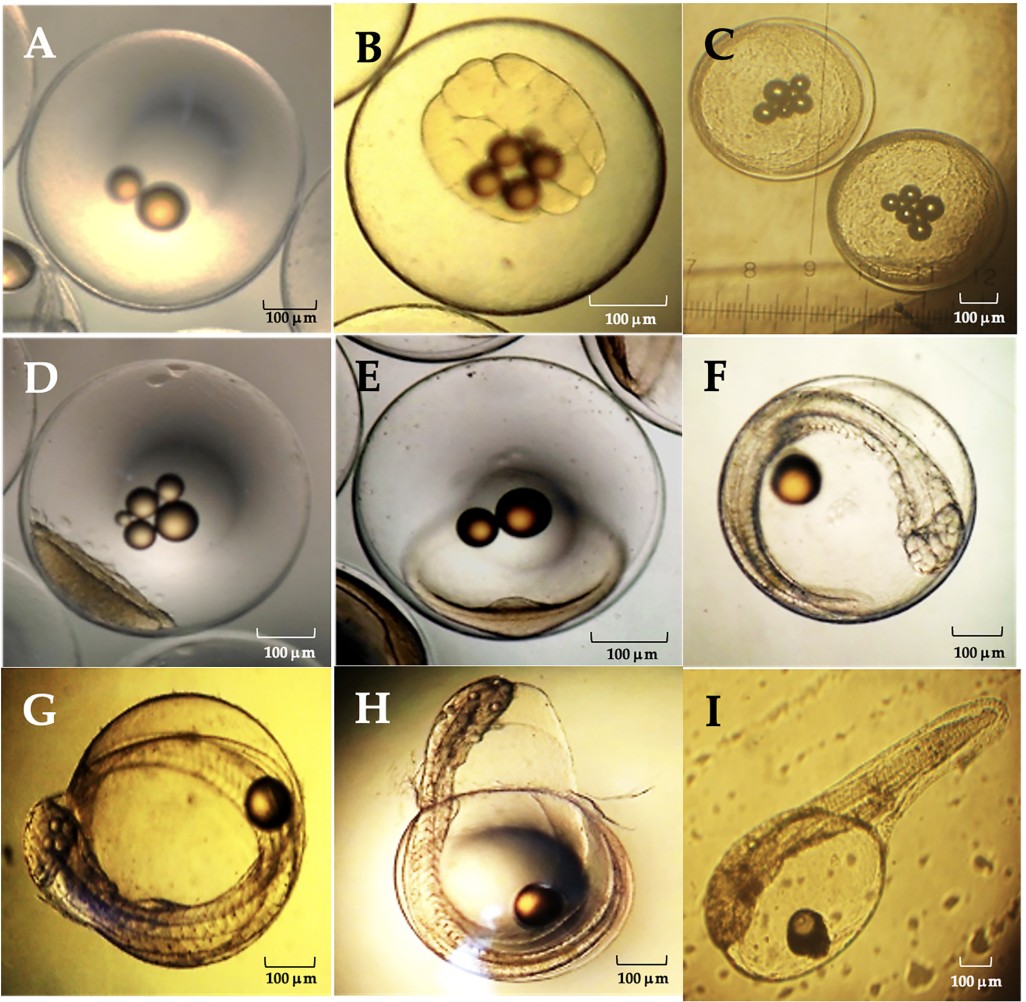

**Figure 10 Relevant stages of the embryonic development of *A. scapularis* (hpf, hours post-fecundation).** (A) Fertilized egg (0.35–0.45 hpf). (B), fifth cleavage (1.00–2.32 hpf; 32 blastomeres). (C) Morula (4.55–5.60 hpf 512-cell stage). (D) Blastula (13.0–14.0 hpf). (E) Gastrula (15.55–18.50 hpf); (F) Organogenesis (20.26–40.00 hpf). (G) Full embryo (41.50–42.15 hpf). (H) Hatching embryo (42.15–42.20 hpf). (I) Hatched larvae (42.25 hpf).

conditions (*Montes et al., 2019*). Second, the length of current hatched larvae was 1.87 ± 0.05 mm at 18 °C but 2.56 ± 0.05 mm at 19 °C (*Montes et al., 2019*). These observations also agree with those of other fishes where the incubation time decreased with the temperature (*Bernal et al., 2001*). Despite that temperature dependence, the thermal tolerance of eggs also depends on the egg's thermal history as well as on compensatory changes in the physiological development (*Jobling & Baardvik, 1994*). Therefore, those key aspects must be weighted when scaling up aquaculture farming of the Peruvian grunt at distinct regional latitudes.

## CONCLUSIONS

A protocol is established for the capture, transport, healthcare and nutrition of wild Peruvian grunt adults to fund an aquaculture broodstock at the onset of its incipient

domestication. The speed of capture and transportation, as well as the preventive treatments, allowed for good survival and a rapid adaptation to captive feeding. The feeding regime based on the copepod *E. analoga* and semi-moist feed allowed for maturation and spawning 6 months after the capture. The knowledge generated on reproduction control and embryonic development contributes towards a regionally based standardization of incubation protocols. The successful scalation of larval production will alleviate fishing pressure on this overharvested species.

## ACKNOWLEDGEMENTS

The authors express their gratitude to Fondo Nacional de Desarrollo Pesquero (FONDEPES) for the rearing facilities provided in the Centro de Acuicultura de Morro Sama (CAMOSA), to the OPI Staff (Oficina de Proxectos Internacionais) from Universidade de Vigo to the OEI Staff (Organización de Estados Iberoamericanos, Educación Superior y Ciencia), as well as to the Technical Staff from FONDEPES for their invaluable technical help during the experimentation. This article was produced within the framework of the SEASOS International Cooperation Network (Euro-Latin Symbiosis for Sustainable Aquaculture) and the Latin American Agroaquaculture Network (SIBIOLAT).

### Funding

The work was funded by the project: "Research and development of farming technologies for marine fish of economic importance: corvina (*Cilus gilberti*) and bream (*Anisotremus scapularis*) in the Tacna region" (the Universidad Nacional Jorge Basadre Grohmann, Rectoral Resolution No. 3780-2014-UN/JBG). The funders had no role in study design, data collection and analysis, decision to publish, or preparation of the manuscript.

### Grant Disclosures

The following grant information was disclosed by the authors:
Research and Development of Farming Technologies for Marine Fish of Economic Importance: Corvina (*Cilus gilberti*) and Bream (*Anisotremus scapularis*) in the Tacna Region.
Universidad Nacional Jorge Basadre Grohmann, Rectoral Resolution:
3780-2014-UN/JBG.

### Competing Interests

The authors declare there are no competing interests.

### Author Contributions

- Renzo Pepe-Victoriano conceived and designed the experiments, performed the experiments, analyzed the data, prepared figures and/or tables, authored or reviewed drafts of the article, and approved the final draft.

- Jordan I. Huanacuni conceived and designed the experiments, performed the experiments, analyzed the data, prepared figures and/or tables, authored or reviewed drafts of the article, and approved the final draft.
- Pablo Presa analyzed the data, authored or reviewed drafts of the article, and approved the final draft.
- Luis Antonio Espinoza-Ramos conceived and designed the experiments, performed the experiments, analyzed the data, authored or reviewed drafts of the article, and approved the final draft.

### Animal Ethics

The following information was supplied relating to ethical approvals (*i.e.*, approving body and any reference numbers):

Director of the Scientific Ethics Committee-AquaInnova Ltda (Res. 009-2024).

### Field Study Permissions

The following information was supplied relating to field study approvals (*i.e.*, approving body and any reference numbers):

The evaluations were carried out at the Morro Sama Aquaculture Center of the Fondo Nacional de Desarrollo Pesquero.

### Data Availability

The raw data is available in the Supplemental Files.

### Supplemental Information

Supplemental information for this article can be found online at http://dx.doi.org/10.7717/peerj.18655#supplemental-information.

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
