# Peer review of "Reproductive management: conditioning, spawning and development of Peruvian grunt Anisotremus scapularis in southern Peru"

_PeerJ, doi:10.7717/peerj.18655_

## Round 0.1 · original submission · Major Revisions

I've received two reviews of your manuscript. It's been a challenge finding reviewers, and many declined to review while others did not reply at all. The reviewers who participated recommend minor and rejection. Consequently, I can give you the opportunity to respond, mainly to the reviewer recommending rejection. This latter argues that "the novelty and significance of the study are not clearly evident, particularly given the reliance on a single treatment and the similarity to previous studies on this species (e.g., Montes et al., 2019)", among many other corrections and suggestions, If you feel you can respond to every and each of the recommendations, go ahead. I will return the revision to these reviewers.

Reviewer 1 ·

Basic reporting

This article describes the methodology for collecting and conditioning wild Peruvian grunt (Anisotremus scapularis) to establish a viable broodstock and study their embryonic development. However, I am unable to recommend acceptance of the current manuscript due to the simplicity of the experiments and the lack of in-depth discussion. Additionally, the novelty and significance of the study are not clearly evident, particularly given the reliance on a single treatment and the similarity to previous studies on this species (e.g., Montes et al., 2019). Furthermore, some of the statistical analyses used appear to be inappropriate for the data presented. Therefore, I do not believe this paper is sufficiently qualified for publication in PeerJ.

Experimental design

For this study, the terminology of embryonic development was used should be refer to the reference cited (who defined fish developmental stages). Since you have only one treatment, using ANOVA to compare growth parameters across months is not appropriate.

Validity of the findings

The authors mentioned that aspects of Montes et al. (2019) were already studied in vitro, specifically regarding the biology of embryonic and larval development in the species. Could you explain this aspect further? Are your findings well integrated with existing knowledge?

Additional comments

Line 48: Add a full stop (.) after Espinoza-Ramos, 2019).
Line 200: 'Floating eggs' should be 'Hatched larvae'.
Line 218: seabream?
Line 226–232: Authors should show the information of gender and size (body weight or body length) of broodstock.
Figure 8: Where did you reference Figure 8 in the manuscript text?
Figure 10 must include the scale bar. Compared to Montes et al. (2019), why does the same species exhibit different embryonic development patterns? For example, before the organogenesis stage, Montes et al. (2019) observed a single oil globule in the fertilized egg, whereas yours has multiple oil globules?

Reviewer 2 ·

Basic reporting

The article provides an overview of the study, identifying the main challenges that must be addressed. To improve readability and technical quality, I offer the following recommendations:

Abstract:
Including the results' relevance to the fishing industry and ecology is recommended. Highlighting how the study contributes to reducing pressure on wild marine populations will demonstrate the practical and environmental significance of the findings.

Introduction:
Although recent literature is cited, a deeper exploration of research related to other marine species' reproduction and domestication strategies would be beneficial. Expanding this theoretical background will better contextualize the strategies used and strengthen the study's relevance within aquaculture.

Experimental design

Materials and Methods:
The methods section clearly describes the capture, transport, acclimatization, feeding, and broodstock management stages. However, providing a more detailed justification for the chosen methodologies is suggested to reinforce the scientific foundation. A comparative analysis with alternative methods would further support the rationale behind these choices.

Validity of the findings

Results:
The results are well-presented with organized tables and graphs. Adjusting the map figure by including a more detailed legend and improving the visualization of the location, particularly in South America, is recommended. Additionally, adding a scale bar to Figure 2 and indicating the standard length of the species is suggested. In Figure 10, reorganizing the legend is necessary, as the numbering in the text does not match the images.
Discussion:
The discussion interprets the results clearly and compares them with similar studies. Offering concrete recommendations on how the findings can be applied in the commercial aquaculture sector will increase their practical impact. Furthermore, considering various environmental and economic factors, expanding the discussion on how these methods could be implemented in other regions is advisable.

Annotated reviews are not available for download in order to protect the identity of reviewers who chose to remain anonymous.

---

## Round 0.2 · Minor Revisions

Two expert reviewers have re-evaluated your revised manuscript. One of the reviewers has some concerns about some issues that were not adequately addressed. Please ensure that you do so in a revised version of the manuscript. Ensure that you clearly note where the changes have been made in your response to reviewers.

Reviewer 1 ·

Basic reporting

The authors have made reasonable efforts in revising and improving the manuscript. But some of my comments were incompletely or inadequately addressed.

Experimental design

Some of my comments were incompletely or inadequately addressed.

Validity of the findings

Some of my comments were incompletely or inadequately addressed.

Additional comments

It is needing minor revision according comments:

2.4. Egg sampling and morphological characterization of the embryo
As I mentioned previously, "For this study, the terminology of embryonic development used should refer to the reference cited (which defined fish developmental stages)." The authors have still not addressed or explained all of my concerns.

Figure 8:
As I mentioned previously, "Since you have only one treatment, using ANOVA to compare growth parameters across months is not appropriate." Why does "Significantly different SGR rates between months are labeled with different superscripts (ANOVA, p < 0.01)" still appear in the figure legend?

Reviewer 2 ·

Basic reporting

The revision of the text resulted in a more concise and fluid wording, correcting problems of clarity and language pointed out in the previous version.

Experimental design

Satisfactory modifications.

Validity of the findings

Satisfactory modifications.

---

## Round 0.3 · accepted · Accept

Thank you for addressing the final reviewer comments. I find the manuscript now suitable for publication in PeerJ.